# Mechanical Properties and Equilibrium Swelling Characteristics of Some Polymer Composites Based on Ethylene Propylene Diene Terpolymer (EPDM) Reinforced with Hemp Fibers

**DOI:** 10.3390/ma15196838

**Published:** 2022-10-01

**Authors:** Maria Daniela Stelescu, Anton Airinei, Alexandra Bargan, Nicusor Fifere, Mihai Georgescu, Maria Sonmez, Mihaela Nituica, Laurentia Alexandrescu, Adriana Stefan

**Affiliations:** 1National Research and Development Institute for Textile and Leather, Leather and Footwear Institute, 93 Ion Minulescu Street, 031215 Bucharest, Romania; 2Petru Poni Institute of Macromolecular Chemistry, 41A Grigore Ghica Voda Alley, 700487 Iasi, Romania; 3National Institute of Aerospace Research “Elie Carafoli”, 220 Iuliu Maniu Blv., 061126 Bucharest, Romania

**Keywords:** EPDM rubber, hemp fiber, crosslink density, mechanical properties, water uptake, reinforcement

## Abstract

EPDM/hemp fiber composites with fiber loading of 0–20 phr were prepared by the blending technique on a laboratory electrically heated roller mill. Test specimens were obtained by vulcanization using a laboratory hydraulic press. The elastomer crosslinking and the chemical modification of the hemp fiber surface were achieved by a radical reaction mechanism initiated by di(tert-butylperoxyisopropyl)benzene. The influence of the fiber loading on the mechanical properties, gel fraction, swelling ratio and crosslink degree was investigated. The gel fraction, crosslink density and rubber–hemp fiber interaction were evaluated based on equilibrium solvent-swelling measurements using the Flory–Rehner relation and Kraus and Lorenz–Park equations. The morphology of the EPDM/hemp fiber composites was analyzed by scanning electron microscopy. The water absorption increases as the hemp fiber loading increases.

## 1. Introduction

Reinforced polymer composites with cellulosic natural fibers have gained increasing importance in recent years, both in the scientific community and in industry. Natural cellulosic fibers have become promising materials to obtain reinforcing agents that can replace inorganic fillers, such as carbon black, silica, etc. Natural fibers offer some advantages, such as easy availability, lower cost, renewable nature, production with less environmental pollution and low health hazards. In addition, the use of natural fibers as a replacement to inorganic fillers is determined by their good mechanical properties, non-toxic nature, non-abrasive behavior on processing equipment and low emission level during incineration at their end of life. Despite these attractive properties, natural fibers used as reinforcing agents have a few disadvantages, such as high level of moisture absorption, lower thermal stability and hydrophilic character, which can lead to incompatibility with hydrophobic polymer matrices. The characteristics of natural fibers depend on climatic conditions, plant species, the harvesting period, the processing conditions, etc. [1,2,3,4,5,6].

Natural fibers are mainly composed of cellulose, lignin, hemicelluloses, pectin, waxes, etc. The chemical composition depends on the type of natural fiber and varies even if they belong to the same family [5,7,8,9]. These compounds, due to their a high content of hydroxyl and polar groups, impart a hydrophilic character to cellulosic natural fibers, leading to poor interfacial interactions with hydrophobic polymer materials. In this way, in order to obtain polymer composites with superior properties, the surface of natural fibers is modified by chemical or physical methods to improve their adherence to the polymer matrix. Chemical treatments such as mercerization, acetylation, maleated coupling, physical treatments (cold plasma treatment, thermal treatment, corona dischargers) or enzymatic procedures were applied to modify the surface of the natural fibers by removing some surface OH-groups [10,11,12,13,14,15]. These methods that treat the surface of natural fibers represent distinct stages in the technological flux to obtain polymer composites reinforced with natural fibers, consequently increasing the working time and the final cost of the composites.

EPDM (ethylene propylene diene terpolymer) rubber has attracted tremendous interest in the industry due to its high resistance to aging, heat, polar solvents and other chemicals, as well as owing to its excellent electrical characteristics and mechanical properties. In addition, EPDM rubber can be processed without difficulties, accepting high levels of fibers, reinforcing agents and plasticizers. The formulation and manufacture of the novel polymer composites based on EPDM and hemp fibers, respecting the requirements of the circular economy, contribute to the diversification of the field of eco-polymer materials and to the improvement of the existing technologies. Due to the advantages of polymer composites reinforced with vegetal fibers, these materials can be used in various applications, such as automotive, aerospace, buildings and construction industries [16,17,18,19,20,21,22], contributing to the reduction in the components’ weight in the final product, improving the damping characteristics to shocks and vibrations of the material [23,24], as well as its thermal and phonic isolation properties [23,24,25]. Furthermore, these composites assure easier processing, higher resistance to extreme temperature variations, and have a low impact on the environment [16,17,18,19].

Hemp (*Cannabis sativa* L.) is an annual plant with a long cultivation history. This plant demonstrates rapid growth, with lower labor requirements and small fertilizer and pesticide dependence. In suitable conditions, the plants can reach 300–1500 mm [4,7,8,9].

In the present work, composites based on EPDM and hemp fibers were obtained. The modification of the hemp fiber surface was performed by a radical reaction mechanism initiated by peroxide, at high temperature during the crosslinking process of the elastomer. The effect of the reinforcing agent loading (hemp fibers) on the physico-mechanical properties, crosslinking density, gel fraction and morphology of the composites was investigated. Moreover, the reaction mechanisms that occur during the crosslinking and grafting process of the EPDM/hemp fiber composites, as well as the characterization of these composites, were discussed.

## 2. Experimental Section

### 2.1. Materials

Ethylene propylene diene terpolymer (EPDM) Nordel 4760 from Dow Chemical Company was used as the polymer matrix. This polymer contains 70 wt % ethylene and 4.9 wt % ethylenenorbornene (ENB), with a Mooney viscosity of 70 ML_1+4_ at 120 °C, density of 0.88 g/cm^3^ and crystallinity degree of 10%. As a lubricating agent, polyethylene glycol (PEG) with medium molecular weight was added. PEG 4000 was supplied by Advance Petrochemicals LTD, with a density of 1.128 g/cm^3^, and melting point in the range of 4–8 °C. It is a flaky solid, easy to process and add to other materials, difficult to precipitate, and has good compatibility with rubber. It can greatly increase the plasticity of rubber products, and effectively reduce the power loss in production processes. In addition, PEG 4000 can contribute to the homogeneous dispersion of the hemp fibers in the EPDM matrix [26,27,28]. A sterically hindered phenolic antioxidant, namely Irganox 1010 (pentaerythritol tetrakis(3-3,5-di-tert-butyl-4-hydroxy)propionate), was added to assure the protection against the thermooxidative degradation and for long-term thermal stabilization. It was bought from BASF Schweiz (melting point of 40 °C, active ingredient of 98% and density of 1.15 g/cm^3^). The reinforcement of the rubber composites was made using ground hemp fiber threads, with a length of 2.5 mm. Perkadox 14-40B di(tert-butylperoxyisopropyl)benzene, used as a vulcanizing agent, was obtained from Akzo Nobel Chemicals (density of 1.60 g/cm^3^, active oxygen content of 3.8%, peroxide content of 40%). In order to determine the crosslink density and the gel fraction, n-hexane (Merck KGaA, Darmstadt, Germany) was utilized as a solvent.

### 2.2. Preparation of EPDM/Hemp Fiber Composites

EPDM-based composites were obtained by the blending method, on a laboratory electrically heated roller mill, equipped with a cooling system (Brabender GmbH&Co KG, Duisburg, Germany) (Figure 1). The working conditions included friction 1:1.1, and temperature of 60–90 °C. The nomenclature of the different samples and the EPDM composite formulations are listed in Table 1. First, EPDM was introduced into the roller mill (2 min), then the antioxidant (Irganox1010) and PEG 4000 were incorporated and mixed for another 2 min. When a homogeneous mixture was obtained, predetermined amounts of ground hemp fibers (0, 5, 10, 15 and 20 phr, respectively) were added and further mixed for 4 min. The sample EP0, without hemp fibers, was considered as the control sample. At the end, the vulcanizing agent was added (1 min) and the mixture was homogenized for 5 min and then removed from the roll in the form of a sheet about 2 mm thick. The vulcanization was carried out using a laboratory hydraulic press (Fortune Press, model TP/600, Fontijine Grotness Vlaardingen, The Netherlands) at 160 °C, pressing force of 300 kN and vulcanization time of 20 min. Subsequently, the specimens were cooled to 45 °C under the pressure force of 300 kN and cooling time of 10 min.

### 2.3. Measurements

The hardness (in Shore A) of EPDM-based composites was determined on a hardness tester according to ISO 48-4 and ASTM D2240, using samples with the thickness of 6 mm. The elasticity was measured on a Schob test instrument on 6 mm thick samples according to ISO 4662 and ASTM D78121-05. The tensile strength, elongation at break and modulus at 100% were measured using dumb-bell shaped specimens according to ISO 37 and ASTM D41 on a Schopper strength tester at a testing speed of 500 mm/min. Equilibrium solvent swelling measurements were taken in hexane at 23–25 °C. The samples were cut in the form of discs with the diameter of 20 mm and thickness of 2 mm. For each sample, three discs were used. The samples were dried at 100 °C for 60 min, then weighed (*m_i_*) and immersed in dark color bottles with lids, in which hexane was introduced. The immersion time was 168 h to attain the equilibrium. After immersion, the samples were reweighed (*m_s_*). Furthermore, the samples were dried at room temperature for 6 days and then at 70 °C for 3 h to remove solvent traces, and finally, the samples were weighed again (*m_d_*). Gel fraction was calculated according to Equation (1) [26,29].
(1)$$Gel\;fraction\left(\%\right)=\frac{m_{d}}{m_{i}}\times 100$$

The crosslink density, ν, was determined using the Flory–Rehner equation for tetrafunctional networks, Equation (2) [30].
(2)$$\nu_{cross}\left(\frac{mol}{g}\right)=\frac{1}{2M_{c}}=-\frac{ln\left(1-V_{r}\right)+V_{r}+\chi_{12}V_{r}^{2}{}_{}}{2V_{s}\left(\sqrt[{3}]{V_{r}}-\frac{V_{r}}{2}\right)}$$
where *M_c_* is the average molecular weight of the rubber between the crosslinks, *V_r_* is the volumetric fraction of rubber at swelling equilibrium, *χ_12_* describes the Flory–Higgins polymer–solvent interaction parameter and *V_s_* is the solvent molar volume (130.77 cm^3^/mol for n-hexane).

The volume fraction of rubber in the swollen sample *V_ro_* for rubber composites that do not contain reinforcing agent can be estimated by the Equation (3) [31,32].
(3)$$V_{r0}=\frac{Volume\;of\;rubber}{\left(Volume\;of\;rubber\right)+\left(Volume\;of\;solvent\right)}=\frac{\frac{m_{d}}{\rho_{r}}}{\frac{m_{d}}{\rho_{r}}+\frac{m_{s}-m_{d}}{\rho_{s}}}$$
where *ρ_s_* represents the solvent density, *ρ_r_* is rubber sample density and (*m_s_−m_d_*) is the weight of solvent in the swollen rubber. The density of the elastomer samples was measured according to ISO 2781.

For the EPDM composites containing reinforcing agent, in the expression of *V_ro_*, the fractions of insoluble compounds must be eliminated and the relation for *V_rf_* becomes Equation (4) [10,33,34,35].
(4)$$V_{rf}=\frac{\frac{m_{d}-\varphi_{h}*m_{i}}{\rho_{r}}}{\frac{m_{d-\varphi_{h}*m_{i}}}{\rho_{r}}+\frac{m_{s}-m_{d}}{\rho_{s}}}$$
where *φ_h_* denotes the volume fraction of hemp fibers, evaluated from Equation (5), which is as follows:(5)$$\varphi_{h}=\frac{\frac{m_{h}}{\rho_{h}}}{\frac{m_{h}}{\rho_{h}}+\frac{m_{r}}{\rho_{r}}}$$
where *m_h_* is the weight of hemp fibers from the sample, *ρ_h_* is the hemp fiber density, *m_r_* is the weight of EPDM rubber and *ρ_r_* is its density.

The Flory–Higgins polymer–solvent interaction parameter *χ_12_* is a measure of the interaction energy of solvent molecules with rubber, and it can be estimated using Equation (6) [36,37,38,39].
(6)$$\chi_{12}=\beta +\frac{V_{s}}{RT}{\left(\sigma_{r}-\sigma_{s}\right)}^{2}$$
where *β* represents the lattice constant for the polymer–solvent system (*β* = 0.34), *R* is the universal constant of gases, *T* is the absolute temperature, *σ_r_* and *σ_s_* describe the solubility parameter of the rubber sample and solvent, respectively [40].

The dynamic water vapor sorption was measured with an IGAsorp dynamic sorption analyzer (Hiden Analytical, Warrington, PA, USA), equipped with an ultrasensitive microbalance, which allows one to observe the modifications in the sample mass as a function of the relative humidity (RH). Before sorption experiments, each sample was dried in nitrogen flow (250 mL/min), until their weight reached the equilibrium at a relative humidity of <1%. The determinations were performed at 25 °C in a RH range of 0–90%, by humidity steps of 10%, with each step consisting of a pre-established equilibrium time between 40 and 60 min.

Fourier transform infrared (FTIR) spectra of all the samples were achieved on a Nicolet IS50 FT-IR spectrometer (ThermoFisher Scientific, Bremen, Germany) in the wavenumber range of 4000–400 cm^−1^, using attenuated total reflection (ATR).

The morphology of the EPDM-based composites was analyzed using a Quanta 250 scanning electron microscope (FEI, Brno, the Czech Republic) on square shaped samples of 10 × 10 mm, with a magnification between 100 and 10.000. For the morphostructural measurements, a conductive layer was deposited on the sample surface.

## 3. Results and Discussion

### 3.1. Reaction Mechanism

The crosslinking of EPDM/hemp fibers, as well as the chemical modification of the hemp fibers, was carried out at 160 °C using Perkadox 14-40B as a crosslinking agent. The crosslinking process takes place in several stages, which are as follows. The initiation stage of the crosslinking reaction occurs in two successive steps (Figure 2). In the first step, the crosslinking agent (Perkadox 14-40B) decomposes at 135–160 °C and the free radicals RO ∙or R are formed. These radicals are active species and they will react with both EPDM elastomer and the cellulose, the main component of the vegetal fiber. In the second step, due to the formed free radicals, hydrogen atom abstraction can occur from the EPDM polymer chains and from polymeric components of the hemp fibers. In the EPDM matrix, the abstraction of hydrogen atoms will occur probably at the –CH_2_—and = CH- units from the polymer main chain and at the allylic positions C_3_ and C_9_ of the diene unit (Figure 2), forming alkyl EPDM and allyl macroradicals [9,41,42,43]. In addition, at the interface between the elastomer matrix and the polymeric components of the hemp fibers, some reactions between the formed radicals and the polymeric components of hemp fibers can occur. It is well-known that the composition of the hemp fibers includes cellulose (70–74%), hemicellulose (15–20%), lignin (3.5–5.7%), pectin (0.8%) and waxes (1.2–6.2%), depending on the type of the natural fiber, soil nature, climate, etc. [7,8,9]. In the above-mentioned reaction mechanism, only cellulose was taken into consideration, because it is the main compound in the hemp structure. The reactivity of cellulose is determined by the presence of the three equatorially positioned OH groups in the anhydroglucopyranose moieties, one primary and two secondary groups [44,45]. In this way, it can be considered that the hydrogen atoms’ abstraction can occur from these positions.

In the second stage, by the addition reaction between the formed stable macroradicals and the polymeric compounds from the system, the appearance of the crosslinking was detected, with the formation of some macroradicals of higher molecular weight (Figure 3). In this case, a three-dimensional network (crosslinked EPDM/hemp) and some structures containing EPDM polymer chains and cellulose or other components from the natural fiber can be formed, which will improve the compatibility between the natural fiber and the elastomer matrix.

In the termination stage, two possibilities exist. In the termination by a combination process, two growing polymer chains react with the mutual decrease in the growth activity. The termination by recombination (alkyl/allyl, allyl/allyl, alkyl/allyl) of two EPDM macroradicals (EPDM) results in a crosslinking reaction (EPDM-EPDM), the reaction of EPDM macroradicals with cellulose macroradicals (cellulose), forming compatibilizing agents (EPDM-Cellulose), or with peroxide radicals, resulting in a crosslinking inactive reaction (Figure 4).

In the hydrogen transfer reactions, a growing polymer chain is deactivated by transferring its growth activity to a previously inactive species. The hydrogen transfer reactions can take place from the ENB allylic positions C3 and C9 to radicals formed earlier in the peroxide curing of EPDM, but without the unsaturation being consumed. At the same time, some scission reactions of the polymer chain can occur simultaneously. The number of these reactions increases as the content of propylene units increases in EPDM rubber [46]. Because the EPDM rubber used here contains a low level of propylene units (25.1%), the number of these reactions will be small and the properties of the vulcanized materials will not be affected.

### 3.2. Mechanical Properties

The mechanical properties have been described by hardness, elasticity, 100% modulus, tensile strength and elongation at break. The hardness of the EPDM/hemp fiber composites increases as the amount of hemp fiber increases, indicating the reinforcing effect of the hemp fibers in EPDM-based composites (Table 2). The maximum hardness was observed for composites loaded by 20 phr hemp fibers (26%). For the other EPDM composites, an increase in hardness of 8% for the EP5Ca sample, 19% for the EP10Ca sample and 23% for the EP15Ca sample, respectively, was recorded. Higher crosslink density in EPDM composites determines higher values of hardness (Table 1 and Table 2).

The elasticity (rebound resilience) shows high values for the sample without hemp fiber and then its values decrease as the content of the hemp fibers increases in EPDM composites (Table 2). The reinforced EPDM composites display a decrease in elasticity of 13% for the EP5Ca sample to 28% for the EP20Ca sample, suggesting that the hemp fibers introduced in the EPDM matrix determine the decrease in elasticity and the segment mobility, in agreement with the results observed for other reinforcement agents [47,48]. The modulus at 100% elongation increases with increasing hemp fiber loading. Thus, an increase of about 27% was obtained for the EPDM composites, due to the fact that the introduction of the reinforcing agent improves the stiffness of the material [49].

The tensile strength of all the composites was reduced with the increasing level of hemp fibers, as compared to the control sample. The decrease in the tensile strength with 5–11% in EPDM composites can be determined by the decrease in the crystallization degree, due to the fiber reinforcement and the increase in the crosslinking degree. In addition, this decrease may be due to an inhomogeneous distribution of the reinforcing agent in the composites, leading to more sites of stress concentration and the decrease in the polymer chain mobility under mechanical loading [49,50,51,52]. Moreover, the values of the elongation at break are high, over 193% compared to the control sample, but the decreasing trend as the hemp fiber content increases was maintained. This decrease may be caused by the restriction of the macromolecular chain movements due to the crosslinking formation [53,54].

### 3.3. Equilibrium Swelling Studies

The equilibrium swelling experiments were used to determine the gel fraction and the crosslink density, as well as the rubber–reinforcing agent interactions, in EPDM composites. The gel fractions were estimated according to relation (1) and in Figure 1, the gel fraction as a function of hemp fiber loading is shown. As can be observed, high values of the fraction gel were obtained over 97%, and a slight increase of 1.12–1.177% was found as the hemp fiber loading increases, reaching 99% for sample EP20Ca as a result of the increase in the reinforcing degree.

The crosslink density was determined according to relations (2)–(6), where the value of the Flory–Higgins polymer–solvent interaction parameter, *χ*_12_, can be estimated. The solubility parameter of hexane as the solvent is 14.9 MPa^1/2^ [39]. The solubility parameter for EPDM rubber can vary in the range of 15.9–18.6 MPa^1/2^ [39], depending on the elastomer composition. In this way, the EPDM solubility parameter (*σ_r_*) was evaluated using the values of molar attraction constants, E, taking into account the structural formula of the polymer and its density, Equation (7) [55,56].
(7)$$\sigma_{r}=\frac{\rho \sum E}{M}$$
where *ρ* and *M* are the density and molecular weight of the polymer repeating unit, respectively. To calculate the parameter *σ_r_*, we used the structural unit of EPDM with the composition taken from 2.1. The molar attraction constants of the polymer repeating unit (Table 3) were taken from the literature [55,57]. The computed value using relation (7) of the solubility parameter for EPDM rubber Nordel 4760 was found to be 16.5 MPa^1/2^. With this value and using relation (6), the value of the Flory–Higgins polymer–solvent parameter, *χ*_12_, was 0.47. The results obtained using the Flory–Rehner equation for the crosslink density are depicted in Figure 2. It can be observed that the crosslink density increases with increasing hemp fiber loading for EPDM composites from 2.64 × 10^−4^ mol/cm^3^ to 4.03 × 10^−4^ mol/cm^3^, suggesting the reinforcement of EPDM composites.

The EPDM matrix reinforcing agent interaction was measured from the equilibrium swelling experiments using the Kraus Equation (8) [58,59].
(8)$$\frac{V_{r0}}{V_{rf}}=1-m\left(\frac{\Phi }{1-\Phi }\right)$$
where *V_ro_* denotes the volume fraction of the equilibrium swollen rubber without filler (given by Equation (3)), *V_rf_* is the volume fraction of the equilibrium swollen rubber with the reinforcing agent (given by Equation (4)), *m* is the polymer–filler interaction parameter and Φ is the volume fraction of the fiber. The Kraus plot of the EPDM composite is shown in Figure 3. The value of m can be obtained from the slope of *V_ro_/V_rf_* as a function of *Φ/(1−Φ).* The interaction constant between rubber and filler, C, can be estimated from Equation (9) [32].
(9)$$m=3C\left(1-V_{r0}^{1/3}\right)+V_{r0}-1$$

Figure 3 illustrates the decrease in the *V_ro_/V_rf_* ratio as the content of the reinforcing agent increases in the composites, indicating the reinforcement effect of the hemp fibers and improved rubber fiber adhesion. The better interfacial bonds between the reinforcing agent and the polymer matrix hinder the solvent penetration in the matrix and the reinforced vulcanizates present lower values of the ratio *V_ro_/V_rf_* [50,60]. These observations are confirmed by the values of constants m and C. The constants m and *C*, determined according to Figure 3, have the following values: *m* = 1.0623 (with a regression coefficient of *R^2^* = 0.9821) and *C* = 2.1235, respectively. These values suggest a good interaction between the hemp fibers and the elastomer matrix [58,59]. In addition, the negative slope of the Kraus plot indicates the reinforcement effect of the hemp fibers in the composites.

The extent of interactions between EPDM rubber and hemp fibers can be analyzed using the Lorenz–Park Equation (10) [32,61,62].
(10)$$\frac{Q_{f}}{Q_{s}}\;=a\;e^{-z}+b$$
where *z* is the weight fraction of fiber in the polymer, *a* and *b* are constants, *Q_f_* and *Q_s_* represent the amount of solvent absorbed per unit weight of rubber and filled material, respectively, and they are obtained using Equation (11) [32].
(11)$$Q=\frac{m_{s}-m_{d}}{m_{d}}$$
where *m_s_* and *m_d_* are the swollen and dried weight of the compound.

It can be observed from the plot of *Q_f_/Q_s_* as a function of *e^−z^* (Lorenz–Park plot) that the ratio *Q_f_/Q_s_* decreases with increasing hemp fiber loading, suggesting an improvement of the interaction between the EPDM rubber and the hemp fiber. The lowest value of the *Q_f_/Q_s_* ratio was obtained for the sample containing 20 phr of the hemp fiber; thus, maximum interaction between the hemp fibers and the EPDM matrix was found for this loading. The values of constants *a* and *b* are characteristic for each system and they can be obtained from the slope and the intercept of the straight line from Figure 4. It can be observed that the value of parameter *a* is twice the value of parameter *b*, making evident the good interaction between rubber and the reinforcing agent. As can be observed from Figure 4, the combination of the high values of *a* with lower values of *b* leads to strong filler–EPDM rubber interactions [32,63], which is in agreement with the results obtained using the Kraus relation.

### 3.4. FTIR Spectra

The FTIR spectra of the EPDM/hemp fibers are presented in Figure 5. The IR spectra of the hemp fibers (Figure 5a) showed absorption bands at 2917 and 2850 cm^−1^, due to the CH stretching vibrations of CH and CH_2_ groups in cellulose and hemicellulose. The absorption band around 3293 cm^−1^ can be assigned to the stretching vibrations of OH groups [64,65]. The absorption bands at 1737 and 1635 cm^−1^ are due to the C=O stretching vibration of carboxylic acid in pectin or ester groups in hemicelluloses and OH bending vibration of bonded water in hemicellulose [64,66]. The absorption band at 1541 cm^−1^ indicates the presence of aromatic rings in lignin. The absorption bands in the range of 1314–1464 cm^−1^ are associated with the deformation vibration of the CH_2_ and CH groups of cellulose and hemicellulose [67]. The absorption band at 1246 cm^−1^ corresponds to the stretching vibrations of C-O of the acetyl group in hemicellulose and lignin, while the absorption bands at 1102–1158 cm^−1^ are due to asymmetric C-O-C stretching vibrations from the polysaccharide components [66,68]. The FTIR spectra of the crosslinked EPDM rubber (sample EP0) and of EPDM/hemp fiber composites are displayed in Figure 5b. In the FTIR spectrum of the crosslinked EPDM rubber, the strong absorption bands at 2915 and 2850 cm^−1^ are due to the asymmetric and symmetric CH stretching vibrations, the absorption bands at around 1464 and 1377 cm^−1^ are attributed to the –CH_2_ scissoring vibrations and to CH bending vibration of the –CH_3_ groups and the absorption band at 720 cm^−1^ corresponds to –CH_2_ rocking vibrations of the ethylene sequences from the polymer backbone [26,69,70]. The FTIR spectra of the EPDM/hemp composites exhibited absorption bands in the range of 650–700 cm^−1^, assigned to out-of-phase -OH bending in cellulose and hemicellulose and absorption bands located between 1027 and 1158 cm^−1^ due to the CC, C-OH, CH ring and side group vibrations that exist in cellulose, hemicellulose or lignin [64,65,66,67,68].

### 3.5. SEM Analysis

SEM micrographs of the cryogenically fractured surface of the composites were recorded. The surface morphology of the samples EP0 and EP20Ca is shown in Figure 6. From the images, it can be observed that the hemp fibers, together with the other components of the composites, are uniformly dispersed in the elastomer matrix. This quasi-uniform distribution led to good physical and mechanical properties. The fractured surface in the analyzed area has a heterogeneous aspect, both wavy areas and evenly distributed areas are identified and the presence of particles in the polymer mass is highlighted.

### 3.6. Water Sorption

The EPDM/ hemp fiber composites were studied for water absorption behavior and the corresponding sorption/desorption isotherms are displayed in Figure 7. These isotherms can be considered as type IV with H2 pores with an irregular array of sizes and shapes, exemplifying the adsorption behavior of mesoporous materials [71]. It was noticed from Figure 7 that the water absorption increases with the hemp fiber content in the EPDM composites and the highest water uptake was 2.02% for sample EP20Ca. It was observed that, for all the samples, the water absorption rate was slower at the beginning of the process and then the water uptake becomes faster, for RH values over 60%. The water uptake of the EPDM/hemp fiber composites increased because of the hydrophilic nature of the hemp fiber, while the polymer matrix is hydrophobic. The increase in the hemp fiber loading leads to the increase in the free OH groups in composites and to the formation of more hydrogen bonds with water molecules, determining the weight gain of the EPDM/hemp fiber composites [6,9]. In this way, good interfacial adhesion between the elastomer rubber and the hemp fibers enables slight penetration of water molecules into the composite, although the crosslinking prevents the rearrangement of polymer chains during the water absorption process, thus creating resistance to water penetration in the composites [72].

To evaluate the specific surface area, the Brunauer–Emmett–Teller (BET) model was utilized (Equation (12)), by modeling the sorption isotherms registered under dynamic conditions [73].
(12)$$W=\frac{W_{m}CRH}{\left(1-RH\right)\left(1-RH+CRH\right)}$$
where *W* represents the sorbed water weight, *RH* is the relative humidity, *W_m_* is the weight of water forming a monolayer and *C* is the sorption constant. The values of absorbed water weight are given in Table 4.

The BET model describes the sorption isotherms up to a relative humidity of 40%, relating to the type of the sorption isotherm and the material type. This method can present the isotherms of type II, but also of I, III and IV. The difference between the order of water sorption capacity and the values obtained for the specific surface can be determined by the nature of the functional groups of the polymer from the composite. The average pore size also influences, in a complex way, the sorption capacity of EPDM samples. The BET results are shown in Table 4.

By applying the Barrett, Joyner and Halenda model (BJH), based on the calculation methods for cylindrical pores, the average pore size, *r_pm_*, was determined using Equations (13) and (14) [74].
(13)$$V_{liq}=\frac{n}{100\rho_{a}}$$
(14)$$r_{pm}=\frac{2V_{liq}}{A}$$
where *V_liq_* is the liquid volume, *n* is the absorption percentage, *ρ_a_* is the phase density and *A* represents the specific surface area. The values of the average pore size and of the specific surface area are given in Table 4.

## 4. Conclusions

For the hemp fiber-reinforced EPDM composites, the crosslinking of the EPDM elastomer and the chemical modification of the natural fiber surface were obtained by a radical reaction mechanism initiated by di(tert-butylperoxyisopropyl) benzene at 160 °C. The increase in the hemp fiber content leads to the increase in hardness, gel fraction and crosslink density, indicating the reinforcing effect of hemp fibers. The gel fraction values were over 97% for all composites, with a slight increase for higher hemp fiber loading. The maximum hardness was observed for EPDM composites with higher reinforcing agent levels (15 and 20 phr), namely 76 °ShA and 78 °ShA. The EPDM rubber-reinforcing agent interaction was analyzed using the Kraus relation and the Lorenz–Park equations, and good hemp fiber–rubber interaction was observed, in agreement with SEM analysis. The water absorption of the EPDM/hemp fibers was dependent on the hemp loading. The highest water sorption was 2.02% for the sample with maximum hemp fiber content (20 phr).

## Data Availability

The data presented in this study are available upon request from the corresponding author.

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
