# Peer review of "Mechanical Properties and Equilibrium Swelling Characteristics of Some Polymer Composites Based on Ethylene Propylene Diene Terpolymer (EPDM) Reinforced with Hemp Fibers"

_materials, 2022, doi:10.3390/ma15196838_

Round 1
Reviewer 1 Report
This manuscript report the hemp fiber reinforced EPDM composites made by the crosslinking of EPDM elastomer and chemical modification of hemp fibers. With the introduction hemp fiber, the reinforcing effect of composite is enhanced. The manuscript is well organized with experimental support and mechanism discussion, but lack of the discussion on the background and design rational of the project. Some concerns are listed below:
1. What is the rational of using EPDM and hemp fibers? Is it possible to replace hemp fibers with other natural/synthetic fibers? What is the application of such EPDM/hemp fiber composites?
2. A scheme showing the entire fabrication process is high suggested. Full name of EPDM needs to be mentioned in the main text. What is the function of PEG4000 and Antioxidant?
3. Instead of showing the value of mechanical properties, stress-strain curves and recycling stretch-release cycles showing the elasticity of composite should be included. It can clearly show the mechanical performance.
4. What is parts to 100 parts rubber? Is it based on weight or volume?
Author Response
Reviewer 1
- What is the rational of using EPDM and hemp fibers? Is it possible to replace hemp fibers with other natural/synthetic fibers? What is the application of such EPDM/hemp fiber composites?
In the last decades, new EPDM based composites in which the inorganic compounds were replaced with agricultural residue biomass (rice husk) or natural fibers such as flax, kenaf, latania or wood fibers were developed [a-f] However, from these composites a limited number was dedicated to EPDM/hemp fiber composites [g,h]. In addition, the mechanical properties of hemp fibers are comparable to those of the glass fibers, but hemp fibers have a lower cost and energy to obtain than the carbon and glass ones [i]. The EPDM/natural hemp fiber composites can have various applications in automotive industry, buildings, constructions as interior car components such as seat backs, parcel shelves, boot linens, front and rear door linens, thermal insulation materials, packaging materials, etc. [k-m]
[a]. Mohanty AK, Vivekanandhan S, Pin J.M, Misra M., Composites from renewable and sustainable resources: Challenges and innovations. Science 2018; 362(6414): 536–42.
[b] G. Chen, A. Gupta, T. H. Mekonnen, Silane-modified wood fiber filled EPDM bio-composites with improved thermomechanical properties, Composites Part A: Applied Science and Manufacturing, 159, 2022, 107029,
[c] Kim IT, Lee KH, Sinha TK, Oh JS. Comparison of ultrasonic-treated rice husk carbon with the conventional carbon black towards improved mechanical properties of their EPDM composites. Carbon Letters 2021;31, 1071–1077.
[d] Stelescu MD, Airinei A, Manaila E, Craciun G, Fifere N, Varganici C. Property correlations for composites based on ethylene propylene diene rubber reinforced with flax fibers. Polym Test 2017; 59, 75–83.
[e] H. Anuar, N.A. Hassan, F. Mohd Fauzey, Compatibilized PP/EPDM-kenaf fibre composite using melt blending method, Adv. Mater. Res. 264/265 (2011) 743-747
[f] M. Nasihatgozar, V. Daghigh, T.E. Lacy Jr., H. Daghigh, K. Nikbin, A. Simoneau, Mechanical characterization of novel latania natural fiber reinforced PP/EPDM composites, Polymer Testing, 56, 2016, 321-328
[g] Wang J, Wu W, Wang W, Zhang J. Preparation and characterization of hemp hurd powder filled SBR and EPDM elastomers. J Polym Res 2011;18, 1023–32.
[h] M. D. Stelescu, E. Manaila, M. Georgescu,.M. Nituica, New materials based on ethylene propylene diene terpolymer and hemp fibers obtained by green reactive processing, Materials 2020, 13, 2067
[i]. A. Shahzad, Hemp fiber and its composites − a review, Journal of Composite Materials, 46, 973-986 (2012)
[k] Layth, M.; Ansari, M.N.M.; Pua, G.; Jawaid, M., Islam, M.S. A review on natural fiber reinforced polymer composite and its applications, Int. J. Polym. Sci. vol. 2015, 2015, 243947
[l] M. M. Davoodi, S. M. Sapuan, D. Ahmad, A. Aidy, A. Khalina, M. Jonoobi, Concept selection of car bumper beam with developed hybrid bio-composite material, Materials & Design, 32, 4857–4865 (2011).
[m]. E. Sassoni, S. Manzi, A. Motori, M. Montecchi, and M. Canti, Novel sustainable hemp-based composites for application in the building industry: physical, thermal and mechanical characterization, Energy and Buildings, 77, 219–226 (2014).
- A scheme showing the entire fabrication process is high suggested. Full name of EPDM needs to be mentioned in the main text. What is the function of PEG4000 and Antioxidant?
A scheme describing the entire fabrication process of the EPDM/hemp composites was inserted in text in the experimental part.
Scheme 1. Schematic illustration for the preparation of the EPDM-based composites
The full name of EPDM: ethylene propylene diene terpolymer (EPDM) was mentioned in text (p 2) and further we used EPDM.
Irganox 1010 is a sterically hindered phenolic antioxidant being a highly effective stabilizer for organic substrates such as plastics, elastomers, synthetic fibers, adhesives, waxes. It protects these materials against termooxidative degradation during processing.
Polyethylene glycol 4000 (PEG 4000) is used in the rubber industry as an additive having a lubricating and cooling effect which can greatly increase the plasticity and the service life of rubber products. It mainly neutralizes the acidity of filler in rubbers, accelerates the vulcanization rate and crosslinking process [26-28]. It can act as a compatibilizing agent between the polymer matrix and the hydrophilic hemp fibers.
|
26 |
M. D. Stelescu, A. Airinei, E. Manaila, G. Craciun, N. Fifere, C. Varganici, D. Pamfil, F. Doroftei, Effects of electron beam irradiation on the mechanical, thermal and surface properties of some EPDM/butyl rubber composites, Polymers, 10, 1206 (2018) |
|
27 |
P. Qu, Y. Gao, G. F. Wu, L. P. Zhang, Nanocomposites of poly (lactic acid) reinforced with cellulose nanofibrils, BioResources, 5, 1811-1823 (2010) |
|
28 |
S. Luo, J. Cao, X. Wang, Investigation of the interfacial compatibility of PEG and thermally modified wood flour/polypropylene composites using the stress relaxation approach BioResources, 8, 2064-2073 (2013) |
- Instead of showing the value of mechanical properties, stress-strain curves and recycling stretch-release cycles showing the elasticity of composite should be included. It can clearly show the mechanical performance.
Thank you for the recommendation. We used a dynamometer Schopper strength tester to determine tensile strength, elongation at break and modulus at 100 %. Unfortunately, this instrument does not permit to record stress-strain curves or recycling stretch-release cycles. We intend to acquire such an instrument.
- What is parts to 100 parts rubber? Is it based on weight or volume?
phr means parts per hundred rubber and it is based on the weight of rubber and the corresponding ingredient.

Reviewer 2 Report
This proposed attempt is good. The following comments are focused on the enhancement of the quality of this work.
1. I strongly recommend to include furthermore justifications on your shortlisted compositional elements of your composites such as fiber, matrices, fillers, etc.
2. From lines 68 to 74, the applications of natural composites are mentioned with references from 16 to 19. I have checked these references, wherein automotive, aerospace, structural applications are covered. Thus, I recommend to add a few more relevant references for shock and vibration-based applications, and better thermal also phonic isolation-based applications.
3. The relevant ISO standards are included in the main text. This is a good approach. I also suggest to additionally provide the equivalent ASTM standards for the same test specimens involved in this investigation. For example: ASTM D412 & ISO 37;
4. The important analytical relationships are imposed in the main text, which is a very appreciable attempt. Please try to impose only relevant literature nearby all the equations.
5. Please maintain the same unit system for the parameters. For example, a few of the parameters are mentioned in "mm" and some of the parameters are mentioned in the unit of "cm". This confusion should be removed.
6. Please enumerate all the compositional elements involved in Equations (1) to (14). I have verified, wherein some of the terms are not explained so the expansion of the compositional elements is mandatory.
7. Line 252 should be unbolded.
8. Generally, the tensile strength of the natural fiber-based composite is lower than other conventional composites. In this manuscript, the current authors are also mentioned the same in Lines 266 to 276. Therefore, this kind of reduction in tensile strength is okay for your focused scope?
9. I strongly recommend to rephrase and improve the conclusion section. Most of the attainments are not included in an acceptable manner. Better provide a detailed view of properties enhancement with respect to the additions of hemp fibers 0, 5, 10, 15, and 20 respectively.
Author Response
Reviewer 2
- I strongly recommend to include furthermore justifications on your shortlisted compositional elements of your composites such as fiber, matrices, fillers, etc.
- Ethylene propylene diene terpolymer (EPDM) Nordel 4760 from Dow Chemical Company was used as polymer matrix. This polymer contains 70 wt % of ethylene and 4,9 wt % of ehtylenenorbornene (ENB), with a Mooney viscosity of 70 ML1+4 at 120 0C, density of 0.88 g/cm3 and crystallinity degree of 10%.
- As lubricating agent polyethylene glycol (PEG) with medium molecular weight was added. PEG 4000 was supplied by Advance Petrochemicals LTD, having density of 1.128 g/cm3, melting point in the range 4-8 0C.
It is a solid flake, easy to process and add in composition, difficult to precipitate and has a good compatibility with rubber. It can greatly increases the plasticity of the rubber products, effectively reduces the power loss in production process. Also, PEG 4000 can contribute to the homogeneous dispersion of the hemp fibers in the EPDM matrix [I-IV].
|
I |
M. D. Stelescu, A. Airinei, E. Manaila, G. Craciun, N. Fifere, C. Varganici, D. Pamfil, F. Doroftei, Effects of electron beam irradiation on the mechanical, thermal and surface properties of some EPDM/butyl rubber composites, Polymers, 10, 1206 (2018) |
|
II |
P. Qu, Y. Gao, G. F. Wu, L. P. Zhang, Nanocomposites of poly (lactic acid) reinforced with cellulose nanofibrils, BioResources, 5, 1811-1823 (2010) |
|
III |
S. Luo, J. Cao, X. Wang, Investigation of the interfacial compatibility of PEG and thermally modified wood flour/polypropylene composites using the stress relaxation approach, BioResources, 8, 2064-2073 (3013) |
|
IV |
S.P. Luo, J.Z. Cao, X. Wang, Properties of PEG/thermally modified wood flour/ polypropylene (PP) composites, For. Stud. China 14 (2012) 307-314. |
- A sterically hindered phenolic antioxidant namely Irganox 1010 (pentaerythritol tetrakis(3-3,5-di-tert-butyl-4-hydroxy)propionate) was added to assure the protection against the thermooxidative degradation and for long-term thermal stabilization. It was bought from BASF Schweiz (melting point of 40 0C, active ingredient of 98% and density of 1.15 g/cm3).
- The reinforcement of the rubber composites was made using ground hemp fiber threads with length of 2.5 mm.
- Perkadox 14-40B di(tert-butylperoxyisopropyl)benzene used as vulcanizing agent was obtained from Akzo Nobel Chemicals (density of 1.60 g/cm3, active oxygen content of 3.8 %, peroxide content of 40%).
- In order to determine the crosslink density and the gel fraction, n-hexane (Merk KGaA, Darmstadt, Germany) was utilized as solvent.
Table 1 becomes:
Table 1. Composite formulations. a
|
Ingredients |
Sample |
||||
|
EP0 |
EP5Ca |
EP10Ca |
EP15Ca |
EP20Ca |
|
|
Polymer matrix, EPDM |
100 |
100 |
100 |
100 |
100 |
|
Reinforcing agent, Hemp |
0 |
5 |
10 |
15 |
20 |
|
Lubricating agent, PEG 4000 |
3 |
3 |
3 |
3 |
3 |
|
Antioxidant, Irganox 1010 |
1 |
1 |
1 |
1 |
1 |
|
Vulcanizing agent, Perkadox 14-40B |
8 |
8 |
8 |
8 |
8 |
a Parts to 100 parts rubber (phr).
- From lines 68 to 74, the applications of natural composites are mentioned with references from 16 to 19. I have checked these references, wherein automotive, aerospace, structural applications are covered. Thus, I recommend to add a few more relevant references for shock and vibration-based applications, and better thermal also phonic isolation-based applications.
We added some references for shock and vibration based applications or thermal phonic isolation based applications.
|
26 |
M. D. Stelescu, A. Airinei, E. Manaila, G. Craciun, N. Fifere, C. Varganici, D. Pamfil, F. Doroftei, Effects of electron beam irradiation on the mechanical, thermal and surface properties of some EPDM/butyl rubber composites, Polymers, 10, 1206 (2018) |
|
23 |
Y. Tao, M. Ren, H. Zhang, T. Peijs, Recent progress in acoustic materials and noise control strategies – A review, Appl. Mater. Today, 24 (2021) 101141. |
|
24 |
A. Santoni, P. Bonfiglio, P. Fausti, C. Marescotti, V. Mazzanti, F. Mollica, F. Pompoli, Improving the sound absorption performance of sustainable thermal insulation materials: natural hemp fibres, Appl. Acoust. 150 (2019) 279–289 |
- The relevant ISO standards are included in the main text. This is a good approach. I also suggest to additionally provide the equivalent ASTM standards for the same test specimens involved in this investigation. For example: ASTM D412 & ISO 37;
We mentioned in text the equivalent ASTM standards utilized in this investigation.
Hardness: ISO 48-4 & ASTM D2240; elasticity: ISO 4662 & ASTM D7121-05;
tensile strength: ISO 37 & ASTM D41.
- The important analytical relationships are imposed in the main text, which is a very appreciable attempt. Please try to impose only relevant literature nearby all the equations
The relevant literature for the equations (1), (7) and (9) from text was added as follows:
Relation (1):
|
29 |
A. Abdel-Hakim, A. A. El-Gamal, M. M. El-Zayat, A. M. Sadek, Effect of novel sunrose based polyfunctional electrical properties of irradiated EPDM, Rad. Phys. Chem., 189, 109729 (2021) |
|
26 |
M. D. Stelescu, A. Airinei, E. Manaila, G. Craciun, N. Fifere, C. Varganici, D. Pamfil, F. Doroftei, Effects of electron beam irradiation on the mechanical, thermal and surface properties of some EPDM/butyl rubber composites, Polymers, 10, 1206 (2018) |
Relation (7) - reference [49] and
- Bakhtiarian, P. J. S. Foot, P. C. Miller Tate, Conductive poly(epichlorohydrin)-polyaniline dodecylbenzenessulfonate in solution, Progr. Rubber Plast, Recycling Technol., 32, 183-200 (2016)
Relation (9) - reference [22]
- Please maintain the same unit system for the parameters. For example, a few of the parameters are mentioned in "mm" and some of the parameters are mentioned in the unit of "cm". This confusion should be removed.
We express all the parameters in mm
P 4 30-150 cm → 300-1500 mm.
- Please enumerate all the compositional elements involved in Equations (1) to (14). I have verified, wherein some of the terms are not explained so the expansion of the compositional elements is mandatory
We verified the compositional elements in the equations (1)-(14) and we give below the significance of the terms included in eq. (4).
The volume fraction of rubber in the swollen sample, Vr, for composite which does not contain reinforcing agent is denoted as Vro and can be estimated by the relation (3) [31, 32]. For the EPDM composites containing reinforcing agent, the volume fraction of rubber at swelling equilibrium is designated by Vrf and in the expression of Vr, the fractions of insoluble compounds must be eliminated and the relation for Vr becomes (4) [33-36].
- Line 252 should be unbolded.
Line 252 was unbolded.
- Generally, the tensile strength of the natural fiber-based composite is lower than other conventional composites. In this manuscript, the current authors are also mentioned the same in Lines 266 to 276. Therefore, this kind of reduction in tensile strength is okay for your focused scope?
The decrease of tensile strength of EPDM.hemp composites was observed for other EPDM composites containing natural fibers (flax) (M. D. Stelescu, A. Airinei, E. Manaila, G. Craciun, N. Fifere, C. Varganici, Polym Testing, 59, 75-83 (2017)). However, hardness and 100% modulus increase in EPDM natural fiber composites. We continue our investigations on composites based on other types of elastomers and natural fibers.
- I strongly recommend to rephrase and improve the conclusion section. Most of the attainments are not included in an acceptable manner. Better provide a detailed view of properties enhancement with respect to the additions of hemp fibers 0, 5, 10, 15, and 20 respectively.
We revised the conclusion chapter.
Conclusions
For the hemp fiber reinforced EPDM composites, the crosslinking of the EPDM elastomer and the chemical modification of the natural fiber surface were obtained by a radical reaction mechanism initiated by di(tert-butylperoxyisopropyl) benzene at 160 0C. The increase of the hemp fiber content leads to the increase of hardness, gel fraction and crosslink density indicating the reinforcing effect of hemp fibers. The gel fraction values were over 97% for all composites with a slight increase for higher hemp fiber loading. The maximum hardness was observed for EPDM composites having higher reinforcing agent levels (15 and 20 phr) namely 76 oShA and 78 oShA. The EPDM rubber-reinforcing agent interaction was analyzed using Kraus relation and Lorenz-Park equations and a good interaction hemp fibers-rubber was observed in agreement with SEM analysis. The water absorption of the EPDM/hemp fibers was dependent on the hemp loading. The highest water sorption was of 2.02 % for sample with high hemp fiber content of 20 phr.

Reviewer 3 Report
In this paper, the authors hemp fiber reinforced EPDM composites with systematically varied fiber loading were prepared. The influence of the fiber content on the mechanical properties, gel fraction, swelling ratio and crosslink degree was investigated. However, the conclusion and explanation obtained are not clear and need more supports. The improvements of the paper are required before considering publication. My suggestions are as follows:
1.) There is no evidence for the existence of crosslinking polymer structure presented in Scheme 1-3. Although indirect measurements (e.g. mechanical and swelling properties) show that it does occur, it should also be supported by structural characterization results.
2.) The Introduction part is unfocused. I recommend swapping the last two chapters while the first two chapter of section 3.1 should also be moved to the introduction.
3.) In Eq. 1 md and mi represents the sample masses after drying at 70°C and at 60 °C, respectively. Is this correct? Why isn't ms (swollen sample) in the counter?
4.) Eq 10: Please enter the values of constants a and b with references.
5.) How the presence of fiber in the polymer matrix influences the crystallization degree of the composites?
6.) According to the water sorption measurements the water uptake was increased with the fiber loading. In this respect the last sentence of this chapter (resistance to water absorption…) is not clear. Please clarify.
Author Response
Reviewer 3
1.) There is no evidence for the existence of crosslinking polymer structure presented in Scheme 1-3. Although indirect measurements (e.g. mechanical and swelling properties) show that it does occur, it should also be supported by structural characterization results.
The absorption bands at 1464cm-1 and 720 cm-1 can be connected with crystallinity and high degree of regularity of the linear backbone structure. But, for the composites containing hemp fibers, the absorption band at 720 cm-1 overlaps with the band corresponding to out-of-plane stretching vibrations of the C–H bonds from the aromatic ring of hemp fibers [Fa-Fd]. In the EPDM/hemp composites the intensity of absorption bands around 1462-1465 cm-1 decreases by hemp incorporation, then a slight decrease occurs and then a clear increase takes place for composites containing 10-15 phr hemp fibers, suggesting an increase of the polymer chain by crosslinking and/or by the incorporation of hemp fibers. However, in order to show the presence of the crosslinked polymer structure, beside the mechanical and swelling determinations, the study of the thermal properties is now in progress. The Tg values, melting enthalpy and crystallization enthalpy will make evident the crosslinking extent.
|
Fa |
Ruth, C.; Asensio, M.S.A.M.; José, M.R.; Marisa, G., Analytical characterization of polymers used in conservation and restoration, by ATR-FTIR spectroscopy. Anal. Bioanal. Chem. 2009, 395, 2081–2096 |
|
Fb |
Stelea, L.; Filip, I.; Lisa, G.; Ichim, M.; Drobota, M.; Sava, C.; Muresan, A., Characterisation of hemp fibres reinforced composites using thermoplastic polymers as matrices, Polymers, 2022, 14, 481. |
|
Fc |
M. NÅ£uică-Vîlsan, A. Meghea, M. Sonmez, M. Georgescu, Dynamically cured hybrid polymer nanocomposite based on polypropylene and EPDM rubber, U.P.B. Sci. Bull., Series B, 77(3), 2015, 165-174. |
|
Fd |
J. Coates, Interpretation of Infrared Spectra, A Practical Approach, Encyclopedia of Analytical Chemistry, R.A. Meyers (Ed.), John Wiley & Sons LTD Chichester, 2000 |
2.) The Introduction part is unfocused. I recommend swapping the last two chapters while the first two chapter of section 3.1 should also be moved to the introduction.
The introduction was revised.
EPDM rubber have attracted a tremendeous interest for industry due to high resistance to aging heat, polar solvent and other chemicals as well as having excellent electrical characteristics and mechanical properties. Besides, EPDM rubber can be processed without difficulties accepting high levels of fibers, reinforcing agents and plasticizers. The formulation and achieving of the novel polymer composites based on EPDM and hemp fibers respecting the requirements of the circular economy contribute to the diversification of the field of eco-polymer materials and to the improvement of the existing technologies. Due to the advantages of the reinforced polymer composites with vegetal fibers, these materials can be used in various applications such as automotive, aerospace, buildings and construction industries [16-22], contributing to the reduction of the weight components of the final product, improving the damping characteristics to shocks and vibrations of the material [23,24], to a better thermal and phonic isolation [23-25]. Also, these composites assure an easier processing, higher resistance to extreme temperature variations, low impact on the environment [16-19].
|
16 |
Koronis, G.; Silva, A.; Fontul, M. Green composites: a review of adequate materials for automotive applications. Composites B: Eng. 2013, 44, 120–127. |
|
18 |
Arockiam, N.J.; Jawaid, M.; Saba, N. Sustainable bio composites for aircraft components. In Sustainable Composites for Aerospace Applications, Jawaid, M.; Thariq M., Eds.; Woodhead Publishing: Cambridge, UK, 2018, pp. 109–123. |
|
19 |
Kopparthy, S.D.S.; Netravali, A.N. Review: Green composites for structural applications. Composites Part C: Open Access 2021, 6, 100169. |
|
20 |
Layth, M.; Ansari, M.N.M.; Pua, G.; Jawaid, M.; Islam, M.S. A Review on natural fiber reinforced polymer composite and its applications, Int. J. Polym. Sci. 2015, 2015, 243947 |
|
21 |
M. M. Davoodi, S. M. Sapuan, D. Ahmad, A. Aidy, A. Khalina, and M. Jonoobi, Concept selection of car bumper beam with developed hybrid bio-composite material, Materials & Design, 32, 4857–4865 (2011). |
|
22 |
E. Sassoni, S. Manzi, A. Motori, M. Montecchi, M. Canti, Novel sustainable hemp-based composites for application in the building industry: physical, thermal and mechanical characterization, Energy and Buildings, 77, 219–226, 2014. |
|
23 |
Y. Tao, M. Ren, H. Zhang, T. Peijs, Recent progress in acoustic materials and noise control strategies – A review, Appl. Mater. Today, 24, 2021, 101141, |
|
24 |
A. Santoni, P. Bonfiglio, P. Fausti, C. Marescotti, V. Mazzanti, F. Mollica, F. Pompoli, Improving the sound absorption performance of sustainable thermal insulation materials: natural hemp fibres, Appl. Acoust. 150 (2019) 279–289, |
|
25 |
W. Zhu, S. Chen, Y. Wang, T. Zhu, Y. Jiang, Sound absorption behavior of polyurethane foam composites with different ethylene propylene diene monomer particles, Arch. Acoust. 43 (2018) 403–411. |
3.) In Eq. 1 md and mi represents the sample masses after drying at 70°C and at 60 °C, respectively. Is this correct? Why isn't ms (swollen sample) in the counter?
In eq (1) mi represents the initial weight of the sample. Then, the sample was immersed in hexane. After immersion the sample was reweighted (ms). md is the weight of the dried sample (six days at room temperature and then 70 oC for 3 h) which does not contain non-crosslinked rubber which was extracted with solvent (hexane).
4.) Eq 10: Please enter the values of constants a and b with references
The values of constants a and b are characteristics for each system and they can be obtained from the slope and intercept of straight line from Fig. 4. As can be seen from Fig. 4, the combination of the high values of a with lower values of b leads to strong filler-EPDM rubber interactions [22, A, B].
|
64 |
A. Das, K. W. Stockel Huber, D. Y. Wang, V. Galiatstos, G. Heinrich, Understanding the reinforcing behavior of expanded clay partides in natural rubber compounds, Soft Matter, 9, 3798-3808 (2013) |
|
64a |
J. Ghosh, S. Hait, S. Ghoraj, D. Mondal, S. Wiessner, A. Das, D. De, Cradle-to-cradle approach to waste tyres and development of silica based green tyre composites, Res. Conservation Recycling, 154, 104629 (2020). |
|
Sample
|
Solvent |
Value a |
Value b |
Reference |
|
Natural rubber / clay nanoparticles Natural rubber natural rubber / expanded clay nanoparticle |
toluene |
1.35 3.21 |
0.338 -2.2 |
[I] |
|
SBS rubber/ SiO2 |
toluene |
1.3123 |
0.06725 |
[II] |
|
Polybutadiene rubber / BioChoice Kraft lignin |
toluene |
4.63 |
-2.7 |
[22] |
5.) How the presence of fiber in the polymer matrix influences the crystallization degree of the composites?
The incorporation of the hemp fibers in the polymer matrix as well as the crosslinking of the sample lead to the decrease of the crystallization degree of the composites.
6.) According to the water sorption measurements the water uptake was increased with the fiber loading. In this respect the last sentence of this chapter (resistance to water absorption…) is not clear. Please clarify.
We revised this sentence.
In this way, a good interfacial adhesion between elastomer rubber and hemp fibers can determine the slight penetration of water molecules in composite, although the crosslinking prevent the rearrangement of polymer chains during water absorption process creating thus a resistance to water penetration in composites.

Round 2
Reviewer 1 Report
The author addressed most of the reviewer's concerns in the revised manuscript. However, the quality of the scheme should be improved. Scheme 1 is in poor quality. Words in the scheme cannot be clearly seen, and they even had red lines.
Author Response
Manuscript ID: materials-1895472
Mechanical properties and equilibrium swelling characteristics of some polymer composites based on ethylene propylene diene terpolymer (EPDM) reinforced with hemp fibers
Dear Editor,
Scheme 1 was improved and inserted in text.
The English language and style were checked and the corresponding corrections were made in text with red or blue color.
The references were verified.
Thank you for your support.
Best regards,
Dr. Anton Airinei
Reviewer 2 Report
Thank you very much for your needful modification.
Author Response

(The authors gave the same response as above.)

Reviewer 3 Report
After checking the changes authors have made, I am pleased to recommend the revised manuscript for publication in its current form.
Author Response

(The authors gave the same response as above.)
